# Investigation of the Compatibility between Warheads and Peptidomimetic Sequences of Protease Inhibitors—A Comprehensive Reactivity and Selectivity Study

**DOI:** 10.3390/ijms24087226

**Published:** 2023-04-13

**Authors:** Patrick Müller, Mergim Meta, Jan Laurenz Meidner, Marvin Schwickert, Jessica Meyr, Kevin Schwickert, Christian Kersten, Collin Zimmer, Stefan Josef Hammerschmidt, Ariane Frey, Albin Lahu, Sergio de la Hoz-Rodríguez, Laura Agost-Beltrán, Santiago Rodríguez, Kira Diemer, Wilhelm Neumann, Florenci V. Gonzàlez, Bernd Engels, Tanja Schirmeister

**Affiliations:** 1Institute of Pharmaceutical and Biomedical Sciences, Johannes Gutenberg University Mainz, Staudinger Weg 5, D-55128 Mainz, Germany; 2Institute of Physical and Theoretical Chemistry, Julius-Maximilians-University of Wuerzburg, Emil-Fischer-Straße 42 Süd, D-97074 Wuerzburg, Germany; 3Departament de Química Inorgànica i Orgànica, Universitat Jaume I, 12080 Castelló de la Pana, Spain

**Keywords:** covalent inhibitors, in vitro study, protease inhibitors, peptidomimetic sequence, warhead, reactivity and selectivity study

## Abstract

Covalent peptidomimetic protease inhibitors have gained a lot of attention in drug development in recent years. They are designed to covalently bind the catalytically active amino acids through electrophilic groups called warheads. Covalent inhibition has an advantage in terms of pharmacodynamic properties but can also bear toxicity risks due to non-selective off-target protein binding. Therefore, the right combination of a reactive warhead with a well-suited peptidomimetic sequence is of great importance. Herein, the selectivities of well-known warheads combined with peptidomimetic sequences suited for five different proteases were investigated, highlighting the impact of both structure parts (warhead and peptidomimetic sequence) for affinity and selectivity. Molecular docking gave insights into the predicted binding modes of the inhibitors inside the binding pockets of the different enzymes. Moreover, the warheads were investigated by NMR and LC-MS reactivity assays against serine/threonine and cysteine nucleophile models, as well as by quantum mechanics simulations.

## 1. Introduction

The human organism expresses about 600 different proteases falling into five different catalytic classes: aspartic, cysteine, metallo, serine and threonine proteases [1,2]. With their ability to catalyze irreversible protein hydrolysis, these members of the degradome manage the functions of many proteins through various mechanisms, such as activating or inactivating, e.g., growth factors, cytokines and other enzymes. As a result, they play an important role in physiological and developmental processes. These include DNA replication, cell proliferation and differentiation, but also tissue remodeling and neuronal outgrowth [3,4]. Due to their essential roles in such vital processes, dysregulation of these proteins causes severe pathologic conditions, such as cancer and neurodegenerative or cardiovascular disorders [5,6]. Furthermore, proteases play a key role in infectious diseases of, for example, parasitic or viral origin. African trypanosomiasis, also called sleeping sickness, and Chagas disease are caused by parasites and are classified as neglected tropical diseases and constitute important health issues in Latin American and Sub-Saharan African countries. For both diseases, proteases have been identified, which are essential for the development of the parasites and the progression of the disease [7,8]. The 2019–2020 coronavirus (SARS-CoV-2) outbreak is the most recent example of a viral disease with global impact and burden. The viral replication and spreading is associated with proteases playing crucial roles in the viral life cycle, turning them into valid targets for the design of new anti-infectives [9,10].

Over the course of time, various protease inhibitors have been discovered either by targeted design or serendipity. Depending on the target binding site and inhibition mechanism, the molecular structures vary significantly. These range from small molecules to macrocyclic drugs and from non-covalent to covalent inhibition types [11,12,13]. Until recently, covalent modifiers which consist of an electrophilic trap (warhead) were controversially discussed as therapeutics due to the possibility of unselective reactions with off-target proteins and associated immunogenicity and toxicity. These compounds are emerging as potential drugs due to various inherent advantages, such as longer residence times and an accompanying lower drug dosage necessary for effective therapy [14]. There are many covalent drugs that have been approved, including some protease inhibitors, such as the proteasome inhibitors bortezomib or carfilzomib, for treatment of multiple myeloma, which inhibit the proteasome’s *β*5-subunit in an irreversible manner, due to the permanent covalent bond to the catalytically active Thr-1. On the other hand, the nitriles saxagliptin and vildagliptin for treatment of type 2 diabetes and the recent first-approved cysteine protease inhibitor nirmatrelvir for treatment of COVID-19 bind covalent-reversibly to their target proteases, due to the decomposition of the (thio)-imidate adduct formed between the inhibitor and the amino acid of the protease, which is preferable since covalent-reversible inhibition leads to a lower risk of haptenization and binding to off-targets [15,16,17].

The binding of such covalent protease inhibitors proceeds in two stages. A peptidic or peptidomimetic recognition sequence is mainly responsible for the non-covalent interactions (first step) with the substrate binding pockets. It mainly determines the selectivity profile of the inhibitor towards the protease of interest, due to polar and non-polar interactions between the peptidic residues and the enzyme sub pockets. In the second step, the reaction between the warhead and an active site amino acid residue leads to the formation of a covalent bond, either reversibly or irreversibly, between the drug and the enzyme. This step mainly determines the affinity of the inhibitor to the target protease [14,18]. However, the warhead must be suitable for the respective nucleophilic amino acid residue in the active site. Depending on the type of nucleophile, different warheads can be used to target thiol or hydroxy groups of amino acid residues. Functional groups, such as *β*-lactams, but also boronic acids, which are all considered hard electrophiles with regard to the HSAB theory, are warheads targeting mainly serine and threonine-based proteases. Unsaturated, vinylogous Michael-acceptor-like structures, which are considered soft electrophiles, preferably react with cysteine proteases [18,19,20,21]. There are also warheads, e.g., ketones, aldehydes and nitriles, that are similarly suitable for serine-, threonine- and cysteine-based proteases [22,23,24,25]. Thus, exchanging the warhead can lead to different reactivity and affinity profiles, and alterations to the peptidomimetic/peptidic sequence may affect the selectivity of an inhibitor.

Within this extensive systematic study, we selected peptidomimetic sequences specifically to ensure a high affinity towards the protease of interest, which will be discussed below. We collected information about different kinds of warheads regarding their electrophilic properties and inhibition mechanisms to obtain a well-balanced assortment to potentially target cysteine and serine-/threonine proteases and combined them with the sequences (Figure 1) [18,21,22,24]. In vitro testing of all inhibitors on every target, first with the suited peptidomimetic sequence with differing warheads for their on-target and afterwards towards the off-target proteases, revealed the impact of the peptidomimetic sequences and the warheads on affinity and selectivity. The results indicate that, depending on the protease, every tested warhead behaved differently. The experimental results were compared with molecular docking results, visualizing putative binding modes in order to achieve a better understanding of the characteristics of the tested compounds.

Additionally, a reactivity study was carried out using model compounds containing the seven different warhead types, which were reacted with hydroxy and thiol model nucleophiles representing serine, threonine and cysteine proteases. Quantum mechanical computations of the reactions between the warheads and model nucleophiles were used to explain the experimental reactivity test data. These data highlight the preference of the warheads for specific active site residues.

To our knowledge, this is the first systematic study of this extent to evaluate the inhibition properties of peptidomimetic inhibitors with different warheads described in the literature, including in vitro testing towards a series of selected proteases, reactivity tests of the warheads in solution with model nucleophiles and in silico studies (docking and quantum mechanics and kinetic simulations) to explain the experimentally obtained data.

For our studies, the urokinase-type plasminogen activator (uPA) was chosen as a serine, the *β*5-subunit of the proteasome as a threonine and human cathepsin S (CatS), SARS-CoV-2 main protease (M^pro^) and *T. brucei* rhodesain (TbCatL) as representatives of cysteine proteases.

The uPA belongs to the trypsin-like serine protease superfamily and contains a catalytic triad consisting of Ser195, His57 and Asp102 [26]. The enzyme is involved in several physiological functions, such as the degradation of the extracellular matrix (ECM), cell migration and thrombolysis [27,28]. Dysregulation of the uPA is involved in the metastasis of several cancer species [29]. We chose Ac–(l)Gly–(l)Thr–(l)Ala–(l)Arg–(warhead) as the specific peptidomimetic sequence for the uPA-inhibitors because of its high selectivity, which has been reported in the literature [30].

The 20S proteasome is responsible for most of the protein degradation in cells but can also lead to cancer by dysfunction [31]. It consists of three *β*-subunits (*β*1, *β*2 and *β*5), each containing a catalytic threonine. Here, we focus on the *β*5-subunit with the catalytic triad Thr1, Lys33 and Asp17, as it has the greatest impact on the proteolytic activity of the 20S proteasome. We selected the peptidomimetic sequence of bortezomib Pyz–(l)Phe–(l)Leu–(warhead) because of its clinically proven properties as a potent drug [32].

As cysteine proteases, we chose CatS, M^pro^ and rhodesain. Since CatS and rhodesain are both members of the papain family, they would allow a closer examination of the selectivity of the tested inhibitors towards related proteases [33]. CatS contains a catalytic dyad consisting of Cys25 and His164 [34]. It is partly tethered at the cell surface and involved in tissue remodeling, which can lead to cancer cell growth and spreading [35]. We utilized the peptidomimetic sequence morpholine–(l)cyAla–(l)Ser(OBn)–(warhead) which has been reported in the literature because of its described affinity and selectivity properties [36].

In contrast to the aforementioned proteases, rhodesain and M^pro^ do not originate from the human organism but play significant roles in the progression of infectious diseases. Rhodesain is essential for the development of the parasite *Trypanosoma brucei rhodesiense*, which is responsible for the sleeping sickness “Human African Trypanosomiasis”. Analogously to CatS, it contains a catalytic dyad consisting of Cys25 and His159 [37]. There are various peptidomimetic sequences that have been published for rhodesain inhibitors. We decided to utilize Cbz–(l)Phe–(l)*h*Phe–(warhead), as it is a commonly used sequence with great affinity and selectivity [38]. M^pro^ originates from SARS-CoV-2 and plays a key role in the virus replication. The active site contains Cys145 and His164 as a catalytic dyad [39]. Similar to the newly published M^pro^ inhibitors, we chose 4-(OMe)-1*H*-indole–(l)Leu–3-[(3*S*)-2-oxopyrrolidin-3-yl]-(l)Ala–(warhead) as the general structure [40]. All peptidomimetic sequences and warheads are illustrated in Figure 1.

**Figure 1 ijms-24-07226-f001:**
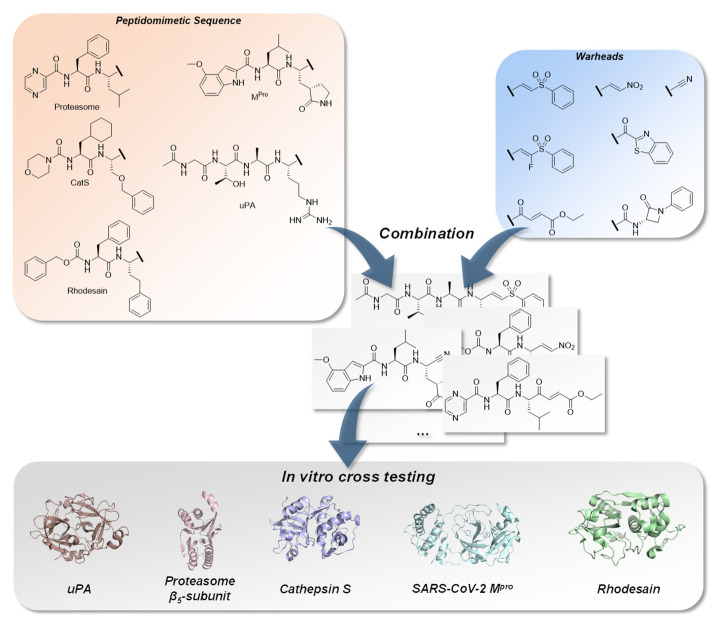
Combination of characteristic peptidomimetic inhibitor sequences for the targets: urokinase-type plasminogen activator (uPA), PDB-ID: 1W10 [41], proteasome *β*5-subunit, PDB-ID: 5LF3 [42], cathepsin S, PDB-ID: 1MS6 [43], SARS-CoV-2 main protease (M^pro^), PDB-ID: 6XR3 [44] and rhodesain, PDB-ID: 2P7U [45], with selected warheads (vinyl sulfone, F-vinyl sulfone, nitroalkene, *α*-ketobenzothiazole, 4-oxoenoate, nitrile and *β*-lactam). The resulting compounds were tested on each target to determine affinity and selectivity.

## 2. Results

### 2.1. Chemistry

#### 2.1.1. Synthesis of Precursors

All tested substances were synthesized in multi-step reactions [19,21]. Regarding the synthesis of the (F-)vinyl sulfone and *β*-lactam compounds, the same precursor molecules were used repeatedly. The preparation of these precursors is shown in Figure 1.

In a substitution reaction on diethyl chlorophosphate (DECP) **2** using methyl phenyl sulfone **1** and *n*-butyllithium (*n*-BuLi), the phosphonate **3** was prepared. Subsequent fluorination of **3** with Selectfluor^®^ led to phosphonate **4**. These precursors were used for the synthesis of vinyl sulfone warheads.

The synthesis of the *β*-lactam precursor **9** was conducted from l-serine. Benzyloxycarbonyl (Cbz) protection followed by amide coupling of the free carboxylic acid moiety with aniline led to the intermediate **7**. The following cyclisation was performed using 1,1′-sulfonyldiimidazol (ImSO_2_) and sodium hydride (NaH). Cbz deprotection with hydrogen and palladium on carbon (Pd/C) yielded precursor **9**.

#### 2.1.2. Rhodesain Inhibitors

The synthesis of substances with the targeting structure designed for rhodesain was conducted according to Figure 2.

The first step of the synthesis of rhodesain inhibitors was the conversion of Boc-(l)*h*Phe-OH **10** into Weinreb amide **11**. From this intermediate, the nitroalkene inhibitor **13** was accessible by reduction to aldehyde **12** and subsequent Henry reaction followed by standard deprotection and amide coupling to connect the P2-P3 residues. In a similar way, the vinyl sulfone **16** and F-vinyl sulfone **17** were obtained, whereby the aldehyde **12** was used in a Horner–Wadsworth–Emmons (HWE) reaction with the precursors **3** and **4** followed by the attachment of the P2-P3 residues. The *α*-ketobenzothiazole inhibitor **18** was prepared by alkylation of the Weinreb amide **11** with benzothiazole and subsequent attachment of the P2-P3 residues. Starting from Boc-(l)*h*Phe-OH, the methyl ester **20** was prepared by amide coupling. A following alkylation with dimethyl methylphosphonate (DMMP) and HWE reaction with ethyl glyoxylate led to the 4-oxoenoate **22**. For the synthesis of *β*-lactam **24**, hydrolysis of methyl ester **20** and amide coupling with precursor **9** yielded the desired product. Nitrile **25** was prepared from carboxylic acid **23** via amide coupling with ammonia followed by dehydration.

#### 2.1.3. Cathepsin S Inhibitors

Compounds designed for the inhibition of cathepsin S were synthesized according to Figure 3.

For the synthesis of cathepsin S inhibitors, the P2–P3 intermediate **29** was used repeatedly. It was prepared by attaching a morpholino-urea residue to cyclohexyl alanine **26** followed by hydrolysis of the methyl ester. In a direct conversion from **29**, the nitrile inhibitor **30** was prepared by amide coupling with ammonia and dehydration. From Boc-(l)Ser(OBn)-OH **31**, the vinyl sulfone **35** and F-vinyl sulfone **36** were obtained by conversion into Weinreb amide **32** followed by reduction, HWE reaction with the precursors **3** and **4** and subsequent standard deprotection and amide coupling with intermediate **29**. Boc-(l)Ser(OBn)-OH **31** was also converted into the methyl ester **37**, which was used for the synthesis of the 4-oxoenoate **40**. Therefore, an alkylation with DMMP and subsequent introduction of the P2 and P3 residues by deprotection and amide coupling led to the phosphonate intermediate **39**, which was converted into the desired product by HWE reaction with ethyl glyoxylate. The *α*-ketobenzothiazole **42** was prepared from methyl ester **37** in an alkylation reaction with benzothiazole and attachment of the P2-P3 residues by deprotection and amide coupling with intermediate **29**. Starting from methyl ester **37***,* deprotection and amide coupling with intermediate **29** led to the methyl ester intermediate **43**, which was converted into the *β*-lactam **45** by hydrolysis and amide coupling with precursor **9**. The nitroalkene **49** also was prepared from methyl ester **43** by firstly converting it to the alcohol **46** and then to aldehyde **47**, which was used in a Henry reaction with nitromethane and subsequent dehydration.

#### 2.1.4. Proteasome β5-Subunit Inhibitors

Compounds designed for the inhibition of the proteasome *β*5-subunit were synthesized according to Figure 4.

The synthesis of proteasome *β*5-subunit targeting compounds started from Boc-(l)Leu-OH **50**, which was converted into the Weinreb amide **51**. From this, the F-vinyl sulfone **55** was prepared by reduction to aldehyde **52** and subsequent HWE reaction followed by a standard deprotection and amide coupling procedure connecting the P2 and P3 residues. For the vinyl sulfone inhibitor **58**, a different route was taken. First, the Weinreb amide intermediate **56** containing the P2 and P3 residues was prepared by standard amide coupling. Subsequent reduction and HWE reaction led to the desired inhibitor. The Weinreb amide **57** was also the intermediate for nitroalkene **59**, which was prepared by reduction and Henry reaction with subsequent dehydration. From Weinreb amide **51**, the *α*-ketobenzothiazole moiety was introduced by alkylation. The attachment of the P2 and P3 residues by standard deprotection and amide coupling yielded the *α*-ketobenzothiazole **62**.

The 4-oxoenoate inhibitor **68** was prepared by HWE reaction of ethyl glyoxylate with the phosphonate intermediate **67**. The latter was synthesized by starting with the Boc protection of H-(l)Leu-OMe · HCl **63**, followed by alkylation of the methyl ester with DMMP and successive deprotection/amide coupling to introduce the P2 and P3 residues. In the same way, the introduction of the P2 and P3 residues to H-(l)Leu-OMe · HCl **63** led to the methyl ester intermediate **70**, from which the *β*-lactam **72** was prepared by hydrolysis and subsequent amide coupling with precursor **9**. Methyl ester **70** was also converted into the nitrile **74** by ammonolysis and dehydration.

#### 2.1.5. SARS-CoV-2 M^pro^ Inhibitors

Compounds designed for the inhibition of SARS-CoV-2 M^pro^ were synthesized according to Figure 5.

Potential SARS-CoV-2 M^pro^ inhibitors were synthesized, starting from the rigidized glutamine analogs **78** and **85**, which had been prepared according to methods reported in the literature [46,47]. The P2–P3 residues fragment of the potential inhibitors was prepared by standard amide coupling with **75** and subsequent deprotection, yielding the intermediate **77**. From glutamine analog **78**, the 4-oxoenoate **81** was prepared by alkylation with DMMP and subsequent deprotection and amide coupling with **77** followed by HWE reaction with ethyl glyoxylate. Also starting from **78**, deprotection and amide coupling with **77** followed by hydrolysis and coupling with ammonia and subsequent dehydration yielded the nitrile inhibitor **84**. Starting with the preparation of Weinreb amide **86** from glutamine analog **85**, the nitroalkene **88** was accessible through reduction, a subsequent Henry reaction with nitromethane followed by dehydration and final deprotection and amide coupling with **77**. Introduction of the *α*-ketobenzothiazole moiety to **86** and connection of the P2–P3 residues by deprotection and coupling with **77** led to *α*-ketobenzothiazole **90**. Similarly, the reduction of **86** and HWE reaction with the precursors **3** and **4** and subsequent attachment of the P2–P3 residues yielded the vinyl sulfone **93** and F-vinyl sulfone **94**. For the *β*-lactam **96**, hydrolysis of **85** and amide coupling with precursor **9** followed by attachment of the P2–P3 residues yielded the desired product.

#### 2.1.6. uPA Inhibitors

Compounds designed for the inhibition of the uPA were synthesized according to Figure 6.

The potential uPa inhibitors are based on a peptide sequence which was synthesized via a standard Fmoc solid-phase peptide synthesis (SPPS) protocol. The obtained peptide **99** was coupled to the *α*-ketobenzothiazole intermediate **102**, which had been prepared from Boc- (l)Arg(Pbf)-OH **100** by alkylation of its Weinreb amide with benzothiazole to yield the *α*-ketobenzothiazole **103**, after deprotection. The vinyl sulfone **106** and F-vinyl sulfone **107** were prepared by reduction of Weinreb amide **101**, followed by a subsequent HWE reaction with the precursors **104** and **105**, which were then coupled with **99** and finally deprotected. The inhibitors with the *β*-lactam, nitrile and 4-oxoeonoate moiety were not synthetically accessible due to the acidic conditions for the Pbf-deprotection to obtain the final inhibitors.

#### 2.1.7. Synthesis of Reactivity Probes

Substances designed for reactivity assay were synthesized according to Figure 7.

For the synthesis of the reactivity probes, leucine was chosen as the model amino acid due to availability and to avoid side-chain reactivity. The different warheads were synthesized in the same way as described above for the full peptidic/peptidomimetic inhibitors. The *β*-lactam **108**, (F-)vinyl sulfone **109**, **110** and nitroalkene **111** reactivity probes were synthesized starting from Boc-(l)Leu-OH **50**, whereas the 4-oxoenoate **112** was prepared from Boc-(l)Leu-OMe **64**. Boc-(l)Leu-1-^13^C-OH **113** was the starting material for the ^13^C-labelled *α*-ketobenzothiazole **115** and nitrile **117** reactivity probes.

### 2.2. Reactivity Tests

To investigate the reactivity between the different warheads towards the three classes of proteases (serine, threonine and cysteine proteases), their behavior in model systems under the same reaction conditions (solvent, nucleophile and base) using either NMR or LC-MS analysis was investigated. We used reactivity probes with a Boc-l-Leu-(warhead) sequence. Leucin was chosen as a P1 amino acid to minimize influences of the side chain and due to synthetic accessibility. 2-Phenylethanethiol was used as a model nucleophile to mimic the thiol moiety of cysteine proteases, and sodium ethoxide was used as a serine/threonine replacement. DMSO-*d*_6_ was used as solvent. Under these conditions, the nucleophile is deprotonated, simulating the activated serine or threonine in the catalytic triad of serine and threonine proteases, while ethanol as protonated alcohol species turned out to be unreactive in preliminary test reactions. The reactivity tests using 2-phenylethanthiol were carried out in the presence and absence of triethylamine as a base. This allowed for a reactivity comparison of the warheads towards protonated and deprotonated nucleophilic thiol species. Generating a deprotonated thiol species in the presence of triethylamine simulates the deprotonated cysteine in the catalytic dyad of cysteine proteases. Figure 8 illustrates the reaction of the reactivity assay with both model nucleophiles and the vinyl sulfone moiety **109** as an example.

The reactivity tests of all Michael acceptors, **109**, **110**, **111** and **112**, the *α*-ketobenzothiazole **115** and nitrile **117** were investigated using an NMR-based analysis method, while the *β*-lactam **108** reactivity was investigated via LC-MS, due to its lack of proton signals, which could be used for evaluation of the reactivity in the ^1^H-NMR studies, and the irreversible reaction mechanism, which allowed the LC-MS analysis. Additionally, LC-MS analyses of all reactions were performed in order to prove the formation of the expected reaction products. Formation of the expected adducts with the nitrile **117** (PhEtSH/PhEtS^–^/EtONa), the *α*-ketobenzothiazole **115** (PhEtSH/PhEtS^–^) and the nitroalkene **111** (EtONa) could not be observed. This may have been due to the covalent reversible reaction mechanism of the nitrile and *α*-ketobenzothiazole and the overall difficult ionization of the specific compounds by an electron spray ionization mass spectrometer.

Method A (NMR): ^1^H-NMR spectra were recorded for the respective warhead and nucleophile mixture, before the addition of the nucleophile (0 min) and after 5, 30, 60, 120 and 240 min reaction time. For quantification, the double bond-signals (doublet/doublet of doublets, around 7.4–6.7 ppm) of the Michael acceptors were integrated relative to 1,3-dioxolane as an internal standard. The acetal CH_2_ signal of the internal standard at 5.3 ppm was used as a reference.

The *α*-ketobenzothiazole **115** and nitrile **117** were similarly analyzed by ^13^C-NMR. Therefore, the corresponding ^13^C-leucin derivates were synthesized (Figure 7). Quantifications of the reactions were carried out by using the integral of the carbonyl carbon atom signal at 195 ppm for the *α*-ketobenzothiazole and 120 ppm for the nitrile moiety. The reference signal of DMSO-*d*_6_ was set to 39.52 ppm.

In Figure 2, the ^1^H-NMR spectra of the test reaction of the 4-oxoenoate **112** with 2-phenylethanthiol **118** are shown exemplarily. After four hours, 92% conversion of the inhibitor to the product **120** was observed. LC-MS analysis confirmed the diastereomeric formation of the expected product **120**.

Method B (LC-MS): The reactivity of the *β*-lactam test compound **108** with the nucleophiles was investigated using an LC-MS-based method. To quantify the conversion, the AUCs were determined at 254 nm. In Figure 3, the UV spectra of the test reaction of the *β*-lactam **108** with EtONa are shown exemplarily. After one hour, complete conversion of the inhibitor to the adduct **122** was observed.

All ^1^H-NMR/^13^C-NMR spectra and chromatograms of the reactivity tests are presented in the Supporting Information (Appendix A). The reactivity test results of all warhead compounds with PhEtSH, PhEtSH + Et_3_N and EtONa are shown in Figure 4.

As depicted in Figure 4A, the 4-oxoenoate **112**, the *α*-ketobenzothiazole **115** and the nitrile **117** moiety did indeed react with PhEtSH under non-basic conditions. In contrast, conversion was not observed with the vinyl sulfone **109**, F-vinyl sulfone **110**, *β*-lactam **108** and nitroalkene **111** warheads. After 240 min, the 4-oxoenoate **112** had nearly completely (92%) reacted with PhEtSH, while only 18% conversion of the *α*-ketobenzothiazole **115** was observed. The equilibrium of the *α*-ketobenzothiazole **116** was reached after 5 min. Similarly, with the nitrile moiety **117**, only 7% conversion was detected, indicating that the formed thioimidate adduct is relatively unstable (Figure 4A). With the addition of Et_3_N (Figure 4B) the overall reactivity increased. Every warhead except the *β*-lactam **108** and the nitrile **117** reacted with the deprotonated thiol species. Full conversion of the 4-oxoenoate **112** could be observed after 5 min, followed by the vinyl sulfone **109**, which took 60 min for complete reaction. The F-vinyl sulfone **110** and nitroalkene **111** both showed similar reactivity with the thiolate species, with a maximum conversion of 86% and 82% after 240 min, respectively. The *α*-ketobenzothiazole **115** also showed an increased reactivity, with around 30% conversion. The reactivity tests with EtONa as nucleophile revealed the 4-oxoenoate **112** moiety as the most reactive warhead, which was completely consumed after 5 min (Figure 4C). However, LC-MS analysis did not prove the formation of the expected product but rather unspecific conversion of **112** (see Appendix A). The nitroalkene **111** showed a similar behavior in comparison to the reactivity test with the deprotonated thiolate species, with a conversion of 84% after 240 min. The *β*-lactam **108** compound showed full conversion after 60 min. In contrast to the deprotonated PhEtSH species, the results indicated a much slower reactivity of the vinyl sulfone **109** with a conversion of 70% after 240 min. No conversion with EtONa was observed for the F-vinyl sulfone **110**. The *α*-ketobenzothiazole **115** showed a higher conversion in the presence of EtONa (37%) than with PhEtSH, but reached this maximum only after 60 min, showing a slower reaction rate compared to the deprotonated thiol species at 5 min. The equilibrium between the *α*-ketobenzothiazole **115** and hemiacetal shifted to 37% conversion and was higher compared to the reactivity test with the thiol nucleophiles. The nitrile **117** showed a similar conversion at 5 min with EtONa compared to the protonated thiol species, with 10% conversion, but again decreased after a period of time, which again indicates the instability of the imidate adduct under basic conditions.

The high reactivity of the 4-oxoenoate **112** warhead with the thiolate is in accordance with the high inhibitory potency of dipeptidyl 4-oxoenoate-based compounds against cysteine protease [48]. The missing reactivity of both vinyl sulfones **109** and **110** toward protonated thiol species and the high reactivity with deprotonated thiols are also in agreement with the high activity of vinyl sulfone inhibitors against cysteine proteases with a thiolate residue in the catalytic center, as reported in the literature.

Nitroalkenes are classified as cysteine targeting warheads, which is also confirmed by the observed reactivity with the model thiolate nucleophile [49].

*β*-lactams are commonly known as warheads in antibacterial agents with transpeptidase-inhibiting properties but have also been used in the development of serine protease inhibitors [21,50,51]. The reactivity tests demonstrate the preference for alcoholate-based nucleophiles, since they only reacted with EtONa and not with PhEtSH/PhEtS^−^.

*α*-Ketobenzothiazole derivatives are used as potent serine and cysteine protease inhibitors [52,53]. Therefore, the reactivity of the *α*-ketobenzothiazole **115** moiety towards all three model nucleophiles was expected. In accordance with the HSAB concept, the stability of the tetrahedral (thio)hemiacetal decreased from the hard sodium ethoxide to the soft thiol/thiolate nucleophiles (EtONa>PhEtS^−^PhEtSH) after 240 min.

The observed reaction of the nitrile **117** with both nucleophiles (PhEtSH/EtONa) is in accordance with the well-known reactivity of nitrile-based drugs. The observed instability of the (thio)-imidate adduct might have been due to the neutral or basic reaction conditions in solution [54,55]. In contrast, the (thio)imidate adduct is stabilized by interaction with amino acid residues of the enzyme pocket [56].

### 2.3. Quantum Mechanics Simulations

As model nucleophiles for the QM simulations, methanethiol/ate and methanolate were used. While the formed products were identical, the warheads vinyl sulfone, F-vinyl sulfone and nitroalkene exhibited varying reactivities for PhEtSH and PhEtSH in the presence of triethylamine. Only for 4-oxoenoate and *α*-ketobenzothiazole was significant reactivity towards PhEtSH observed, whereas for PhEtSH + Et_3_N, all warheads except the nitrile and the *β*-lactam showed reactivity (Figure 4). Since most of the reaction energies with both MeSH and MeSH + Et_3_N were computed to be exergonic (Figure 5A), this cannot be explained merely by thermodynamics. For instance, the experimental results do not show reactivity of the warheads vinyl sulfone, F-vinyl sulfone and nitroalkene with PhEtSH, despite a computed negative free energy of reaction. Thus, to determine whether a reaction can be expected to take place, it is important to consider the whole reaction path, including the activation barriers, which determine the kinetics. Previous calculations have revealed that MeSH is often insufficiently nucleophilic to allow a reaction to occur at room temperature [57]. A base, such as triethylamine, serves as interim storage for the thiol proton before it is transferred to the warhead. By deprotonating the thiol prior to the nucleophilic attack, the nucleophilicity of MeSH is strongly increased, thereby decreasing the associated activation barriers considerably (Appendix A). Following the addition of the nucleophile, the proton is transferred back from the base to the anionic intermediate. Unlike the 4-oxoenoate, vinyl sulfone, F-vinyl sulfone and nitroalkene warheads, the *β*-lactam warhead does not show any reactivity with PhEtSH + Et_3_N. The computed reaction mechanism revealed three consecutive steps to obtain the product (Figure 5C). First, the nucleophile attacks the amide carbonyl group (TS1), resulting in a tetrahedral anionic intermediate (Int1). The rate-determining step is the opening of the lactam ring in the second step (TS2). This was computed to be about 33 kcal mol^−1^ for MeSH + NEt_3_, which is in excellent agreement with the experimental data. In the last step, the former amide nitrogen is protonated by the base to yield the final product (TS3). For the nitrile warhead, a weak reaction with PhEtSH but none with PhEtSH + triethylamine was observed experimentally, which cannot be explained by reference to the computational data. As described in the reactivity tests, this might have been due to the instability of thioimidates in basic solution.

As a result of our calculations, the difference in reactivity between PhEtSH and PhEtSH + Et_3_N for the vinyl sulfone, F-vinyl sulfone and nitroalkene warheads was attributed to a significant reduction in activation barriers caused by proton transfer from the nucleophile to the base prior to the nucleophilic attack. We therefore investigated the reason for the reactivity of 4-oxoenoate and *α*-ketobenzothiazole warheads with PhEtSH in DMSO in the absence of a base. For 4-oxoenoate, a conversion of 92% was observed experimentally, which corresponds to a computed free energy of reaction of about –9 kcal mol^−1^ for the nucleophilic attack at C*_α_* and about –11 kcal mol^−1^ for the addition at C*_β_*_._ The reaction of PhEtSH with *α*-ketobenzothiazole, however, showed only about 18% conversion, and the corresponding product was computed to be 8 kcal mol^−1^ (Figure 5B) and 9 kcal mol^−1^ for the thermodynamic calculation with a bigger basis set (Figure 5A). The solvent used in the experiments was not completely free of water, and, as a result, water molecules were able to catalyze the nucleophilic attack for *α*-ketobenzothiazole and 4-oxoenoate, as well as the keto-enol tautomerization for the latter (Figure 5B and Appendix A) [58]. Our calculations demonstrate that traces of water in the solvent can function as a base to catalyze the reaction of MeSH with the warhead. The activation barrier for *α*-ketobenzothiazole is reduced from more than 40 kcal mol^−1^ to roughly 25 kcal mol^−1^, and the product energy is lowered to 1 kcal mol^−1^ (Figure 5B). Similarly, water catalyzes both the nucleophilic attack of MeSH at the 4-oxoenoate warhead and the subsequent keto-enol tautomerization, leading to a decreased activation barrier of 26 kcal mol^−1^ for the first step (TS1) and one of 20–25 kcal mol^−1^ for the second step (TS2). Additionally, the product energy is even more exergonic at –17-(–18) kcal mol^−1^ (Appendix A). Contrary to the reaction without water, the proton does not have to be transferred directly from the thiol to the atom to be protonated. Instead, it is shuffled along a chain of water molecules. The keto-enol tautomerization is favored for the C*_β_*-addition, but the barrier associated with the rate-determining nucleophilic attack is nearly identical (Appendix A). Thus, it is expected that both reactions should occur in solution. For the reaction with an enzyme, the conformation of the binding pocket will likely determine at which carbon atom the nucleophilic attack will occur.

To mimic the reaction of the warheads with NaOEt, we calculated the reaction path with MeO^–^ and included three water molecules to allow for protonation of the intermediates to obtain the final products and stabilize the reactive anionic species (Figure 5C, Appendix A). The reaction can either terminate at the anionic intermediate or proceed to the neutral adduct by transferring one proton from a water molecule, depending on the basicity of the intermediate, i.e., the intermediate carbanion is poorly stabilized for the vinyl sulfone, hence the reaction progresses to form the neutral addition product (Appendix A). The nitroalkene carbanion, however, is strongly stabilized, and our calculations suggest that the reaction might stop at the intermediate (Appendix A). Analogously, the *α*-ketobenzothiazole forms a deprotonated hemiacetal (Appendix A). Experimentally, no reactivity of the F-vinyl sulfone warhead with NaOEt was observed, which was not supported by our calculations and is contradictory to chemical intuition (Appendix A). As previously stated, the barrier for the *β*-lactam ring opening in reaction with MeSH + Et_3_N was computed to be over 30 kcal mol^−1^, explaining the lack of reactivity in the experiments. Since the anionic species and ring opening are better stabilized in the reaction with MeO^–^ + 3H_2_O, only 23 kcal mol^−1^ is required in this step, which is consistent with the experimental data (Figure 5C).

### 2.4. In Vitro Evaluation of the Synthesized Compounds

Inhibition of the target enzymes was tested via fluorometric assays. Therefore, fluorogenic AMC- or FRET-based substrates with appropriate peptide sequences for the different proteases were used (see Appendix A).

The potential inhibitors were initially screened against all five target enzymes at 20 µM. A cut-off value of 80% inhibition at this concentration was set to differentiate the non-active (n.a.) compounds from active ones.

For the reversible inhibitors (*α*-ketobenzothiazole, nitroalkene, F-vinyl sulfone and nitrile), the IC_50_ values were determined and converted to corresponding *K*_i_ values using the Cheng–Prusoff equation [56]. Regarding the irreversible inhibitors (vinyl sulfone, 4-oxoenoate and *β*-lactam) the *K*_i_, *k*_inact_ and *k*_2nd_ values were determined (see Appendix A) [56]. For a better overview, the p*K*_i_ values were calculated and are presented in Figure 6.

In the following, the inhibition data will be analyzed for each enzyme, first with their suited peptidomimetic sequences (Figure 6, parts A, B, C, D and E), followed by cross testing against the other enzymes.

**uPA**. Only the *α*-ketobenzothiazole inhibitor **103** was found to be active. The combination of the appropriate sequence for uPA with the *α*-ketobenzothiazole warhead resulted in a potent inhibitor with a p*K*_i_ value of 6.9. Other enzymes were not inhibited (Figure 6A).

**Proteasome *β*5-subunit**. None of the compounds with the Pyz–(l)Phe–(l)Leu sequence (**55**, **85**, **59**, **62**, **68**, **72** and **74**), which is well-known from the potent boronic acid-based inhibitor bortezomib, showed inhibition of the proteasome at 20 µM, independently of the warhead used (Figure 6B). Moreover, none of the other compounds with any of the other peptidomimetic sequences showed any inhibition. This highlights the general difficulty of addressing this protease with peptidomimetic inhibitors [56]. An alternative warhead which reacts preferably with Ser or Thr proteases is the epoxide functionality, which is also present in the approved proteasome inhibitor carfilzomib. Although very potent, due to its unpredictable reaction mechanism, this warhead was not included in this study [60].

**CatS**. Regarding the in vitro testing of the cysteine protease CatS, a total of 20 hits were detected (Figure 6C). The most potent inhibitors with the fitting CatS sequence were the nitrile **30** (p*K*_i_ = 9) and the vinyl sulfone **35** (p*K*_i_ = 8.5). The nitroalkene **49** (p*K*_i_ = 7.9) also showed high affinity towards CatS, followed by the 4-oxoenoate **40** (p*K*_i_ = 7.7), the *α*-ketobenzothiazole **42** (p*K*_i_ = 6.5), *β*-lactam **45** (p*K*_i_ = 6.1) and F-vinyl sulfone **36** (p*K*_i_ = 5.5). Since CatS and rhodesain are both papain-like cysteine proteases with similar active sites, cross reactivity between these two series was expected and has been well described in the literature [61]. The vinyl sulfone with the rhodesain-targeting sequence **16** (p*K*_i_ = 9) showed the same inhibition constant as the corresponding inhibitor with the CatS sequence. The vinyl sulfones with the proteasome and the M^Pro^ sequences inhibited CatS to lower degrees (p*K*_i_ = 6.9 and 5.5). A comparison of the 4-oxoenoates of the CatS and rhodesain series yielded the same results, since both exhibited the same p*K*_i_ value of 7.7 for inhibition of CatS. The 4-oxoenoates designed for targeting the proteasome **68** and the M^Pro^
**81** were essentially inactive against CatS (no inhibition in the initial screening at 20 µM). This can be explained by the instability of these compounds in the CatS assay buffer containing dithiothreitol (DTT).

The F-vinyl sulfones, which are reversibly reacting counterparts of the vinyl sulfones, inhibited CatS to a lower degree, and exchange of the peptidomimetic sequence (**36**, p*K*_i_ = 5.5 vs. **17**, p*K*_i_ = 6 vs. **55**, p*K*_i_ = 6 vs. **94**, p*K*_i_ = 5.9) had little to no effect, except for the compounds with the uPA sequence (**103**, **106** and **107**), which was not active at 20 µM against CatS.

The nitroalkene inhibitor which contains the CatS sequence showed a high on-target affinity but changing the sequence to any of the other targeting sequences led to less potent inhibitors (**13**, p*K*_i_ = 6.7 vs. **88**, p*K*_i_ = 6.4 µM vs. **59**, p*K*_i_ = 6.2). Interestingly, the *α*-ketobenzothiazole- (**42**, p*K*_i_ = 6.6) and *β*-lactam- (**45**, p*K*_i_ = 6.1) based inhibitors showed only significant inhibition of CatS if connected to the respective CatS sequence, indicating the strong dependency of a suitable peptidomimetic sequence combined with one of these warheads.

**M^pro^**. In comparison to the M^pro^ inhibitors *α*-ketobenzothiazole **90** (p*K*_i_ = 7.6) and nitrile **84** (*K*_i_ = 7.5) described in the literature, vinyl sulfone **93** (p*K*_i_ = 5.5), F-vinyl sulfone **94** (p*K*_i_ = 6.3) and nitroalkene **88** (p*K*_i_ = 5.7), all of which contain the appropriate M^pro^ peptidic sequence, showed weaker inhibition (Figure 6D) [40,62]. A clear preference of the protease for specific warheads could be observed. The vinylogous warheads (vinyl sulfone, F-vinyl sulfone and nitroalkene) showed significantly weaker inhibition than the *α*-ketobenzothiazole- and nitrile-based compounds. As also observed in the model reactivity studies, the in vitro studies with CatS and both 4-oxoenoate inhibitors with the M^pro^
**81** and the proteasome *β*5-subunit **68** sequences revealed instability in the buffer with DTT. The *β*-lactam **96** as well as all compounds containing a targeting structure designed for other proteases were inactive at 20 µM. This indicates a high specificity of the M^pro^ towards its peptidomimetic sequence.

**Rhodesain**. The results showed similar trends to those found for CatS, with 20 compounds active in the assays (Figure 6E). The most potent was the nitroalkene **13** which contains the corresponding rhodesain peptidic sequence (p*K*_i_ = 10.2), followed by the vinyl sulfone **16**, 4-oxoenoate **22** and nitrile **25**, which showed similar inhibition constants (p*K*_i_-= 7.1–7.7). The F-vinyl sulfone **17** showed moderate inhibition (p*K*_i_ = 5.4), and the *β*-lactam **24** and *α*-ketobenzothiazole **18** were inactive at 20 µM, indicating a preference of rhodesain for vinylogous warheads. Comparable to the CatS study, inhibitors lacked selectivity between rhodesain and CatS due to the structural similarity of the proteases. This is evident through the high p*K*_i_ values of the synthesized CatS inhibitors with 4-oxoenoate- **40** (p*K*_i_ = 7.5), nitroalkene- **49** (p*K*_i_ = 8.7), vinyl sulfone- **35** (p*K*_i_ = 7.7) and nitrile- **30** (p*K*_i_ = 7.7) moieties. Surprisingly, the *α*-ketobenzothiazole **42** designed for targeting CatS showed significant inhibition (p*K*_i_ = 6.5), whereas the *α*-ketobenzothiazole with the rhodesain peptidic sequence **18** was inactive. Among the compounds designed for M^pro^ and the proteasome, the nitroalkene derivatives **88** and **59** showed the same potency as the CatS analogue **49**, both with a p*K*_i_ value of 8.7. Interestingly, the vinyl sulfones with the M^pro^
**93** (p*K*_i_ = 5.7) and the proteasome sequence **58** (p*K*_i_ = 6.5) showed significantly lower affinity compared to the vinyl sulfone designed for rhodesain **16** (p*K*_i_ = 4.7). Differently, the F-vinyl sulfones **55** (p*K*_i_ = 6.3) and **94** (p*K*_i_ = 7.7) showed higher affinities than the analogue with the rhodesain sequence **17** (p*K*_i_ = 5.4). The 4-oxoenoate inhibitor **68** (p*K*_i_-= 8.0) designed for the proteasome and the one designed for the M^pro^
**81** (p*K*_i_ = 7.0) also showed strong inhibition. All other inhibitors with the *β*-lactam and *α*-ketobenzothiazole moiety were inactive, as well as the compounds containing the uPA sequence (**103**, **106** and **107**).

### 2.5. Molecular Docking

To further elucidate the impact of the different warhead types on the binding modes of the inhibitors, protease–inhibitor complexes were investigated with non-covalent and covalent docking [63,64]. For the non-covalent docking, special emphasis was laid on the distances between the reactive nucleophilic carbon atoms of the corresponding warheads to the thiol(ate) or hydroxyl(ate) side chains of the cysteine/serine(threonine) active site amino acids, respectively, as estimates for covalent-bond-formation likeliness. Additionally, the impact of the different warhead moieties on the binding conformation of the inhibitors with otherwise identical peptidomimetic recognition sequences was analyzed. The covalent docking setup was used to investigate whether realistic poses for the covalent complexes could be generated and whether larger conformational rearrangements of the ligand may occur after the covalent reaction.

Conventional non-covalent docking yielded generally reasonable binding modes for all complexes resembling interactions of the crystallographic reference ligands and peptidomimetic recognition sequences in their expected subpockets. Additionally, electrophilic warheads were regularly found in close proximity to the nucleophilic catalytic amino acids (Figure 7, Appendix A). 

The docking with rhodesain (pdb entry 2P7U) indicated that the introduction of a Michael-acceptor system as the warhead led to binding poses similar to the co-crystallized reference ligand, with all the essential interactions between inhibitor and enzyme being nearly identical, as exemplified for the docking poses of the nitroalkene inhibitor **13** (Figure 7a). The poses of the covalent and the non-covalent docking showed that the overall orientation of the inhibitor inside the active site should not change much after the covalent reaction, since the final covalent enzyme–inhibitor complex is very similar to the non-covalent complex (Figure 7a). The corresponding electrophilic C-atoms of all warheads were predicted to be in close proximity to the sulfur atom of Cys25 (2.54–3.50 Å), suggesting a high probability for a nucleophilic attack. High docking scores were also found for the nitroalkene inhibitor **13** (FlexX score: –24.03 kJ/mol; MOE score: –2.66), indicating that it should form very favorable non-covalent interactions while correctly placing the electrophilic warhead (distance to Cys25 sulfur: 3.50 Å). This is consistent with the in vitro data, showing that the nitroalkene moiety represents the most potent inhibitor class for rhodesain. Since the *α*-ketobenzothiazole designed for CatS (**42**) surprisingly inhibited rhodesain with a submicromolar affinity, we compared the non-covalent docking poses between **42** and the ketobenzothiazole with the rhodesain sequence (**18**) (Appendix A). Superposition of the non-covalent docking poses showed that both inhibitors have almost the same positioning with the warhead close to Cys-25 (2.5 Å) inside the active site of rhodesain, indicating that both compounds should have similar affinities towards rhodesain. This makes it hard to explain why inhibitor **42** had a significantly higher affinity for rhodesain in the in vitro testing. Since molecular docking is an inaccurate method, flawed docking poses are no rarity. The non-covalent docking method used in this case might not be suited to explaining this in vitro result. The results of the docking with CatS (1MS6) showed similar trends, since the distances between the electrophilic C-atoms of the warheads and the sulfur atom of Cys25 were again in close proximity in all cases (2.69–3.37 Å). The vinyl sulfone **35** and the nitrile **25** had high scores (FlexX score: –27.35/–26.22 kJ/mol; MOE score: –5.32/–3.00 kcal/mol) combined with similar binding geometries for the covalent and the non-covalent docking poses (shown for nitrile inhibitor **25**, Figure 7b). These data are in accordance with the in vitro data showing that the nitrile warhead was the most potent one, but other warheads also led to productive enzyme inhibition.

For SARS-CoV-2 M^pro^ (6XR3), the distances between the electrophilic C-atoms and the Cys145 sulfur atom were overall slightly higher (2.90–4.91 Å) compared to the papain-like cysteine proteases. The *α*-ketobenzothiazole warhead seems to have a very favorable positioning in the binding pocket, as illustrated by the close proximity (3.41 Å) of the electrophilic C-atom to the thiol of the enzyme Cys145 (Figure 7c). Superposition of the covalent and non-covalent docking poses of **90** showed almost identical positioning of the inhibitor inside the enzyme, with most of the polar interactions retained.

Out of all the investigated warheads in this series, only the nitrile, the *α*-ketobenzothiazole and the *β*-lactam warheads are known to react with oxygen containing amino acid residues in serine (uPA) or threonine (proteasome) proteases.

For the uPA, which was the only target with only one hit in the enzymatic assay, non-covalent docking revealed a large distance between the electrophilic C-atom and the hydroxy-group in the active site for the *β*-lactam (5.07Å) as a known serine warhead. Only the *α*-ketobenzothiazole inhibitor **103**, which had one of the highest scores out of all the inhibitors (FlexX score: –51.59 kJ/mol), was in close proximity to the oxygen of Ser195 (2.84 Å distance to the electrophilic C-atom). This inhibitor also showed a high potency in the in vitro study (Figure 7d). Finally, docking of the *β*-lactam containing inhibitor **72** designed for the proteasome revealed that the warhead position was, again, too far away from the threonine oxygen (4.95 Å), possibly preventing a covalent reaction (Figure 7e). This could be explained by the shifted positioning of the lactam moiety compared to the other warheads. Although the docking of the nitrile and *α*-ketobenzothiazole inhibitors **74** and **62** might suggest that these compounds should inhibit their target sufficiently since the warheads are positioned correctly and in close proximity (2.27 Å/3.16 Å) to the Thr-1 oxygen atom, there was still no inhibition with these warheads in the in vitro study. This might have been due to wrongly generated binding poses, since docking approaches are not always reliable and cannot be considered flawless in all cases. A possible explanation why none of the compounds designed to address the *β*5-subunit of the proteasome showed any inhibition might be the catalytic dyad in the active site consisting of Lys33 and Thr1 compared to the catalytic dyads or even triads in the other enzymes, where the deprotonation of the active site residue is assisted by histidine and/or asparagine. The lysine residue might not always be able to deprotonate the threonine in the active site, depending on the inhibitor, and thus facilitate the covalent reaction step with a warhead [65].

### 2.6. Comparison of the Reactivity Assay Results with the In Vitro Study

Based on the reactivity assay, all Michael acceptors (4-oxoenoate **112**, (F-) vinyl sulfone **109**/**110** and nitroalkene **111**) showed high reactivity toward the deprotonated cysteine model nucleophile, which is congruent with the observed behavior of the synthesized compounds designed for CatS and rhodesain inhibition in the in vitro studies. Furthermore, the *α*-ketobenzothiazole warhead **115** showed a strong reactivity for both model nucleophiles (PhEtS^−^/EtONa), which is consistent with the correspondent uPA and M^pro^ inhibitors **103** and **90** in the protease assays. However, the nitrile **117** showed no reaction with the deprotonated cysteine but with the serine model nucleophile, which contradicts the high inhibitory activity against the cysteine proteases and the missing inhibition by the proteasome *β*5-subunit inhibitor **74**. This might have been due to the aforementioned instability of the thioimidate adduct in basic conditions compared to the stabilized adduct in the enzyme pocket and the overall difficulty of addressing the proteasome *β*5-subunit. The *β*-lactam **108** showed only a strong reactivity towards the serine model nucleophile, but the corresponding bortezomib derivative **72** did not inhibit the proteasome *β*5-subunit, which might have been due to the shift of the electrophilic center of the *β*-lactam moiety into the S1’ pocket and the resulting increase in distance. The 4-oxoenoate moiety **112** was the only warhead that showed high reactivity toward the protonated cysteine model nucleophile, which might hint at non-selective reactivity behavior toward thiol species under physiological conditions. This could also be observed in the in vitro studies. The 4-oxoenoate compounds designed for the M^pro^
**81** and proteasome-*β*5-subunit **68** both reacted quickly with DTT in the respective buffer solutions and appeared to be inactive.

## 3. Discussion

Covalent targeting has become a popular and powerful concept in drug discovery, and great efforts have been devoted to developing and repurposing different warheads [66]. In this first extensive systematic study, we aimed to achieve a deeper insight into the reactivities and selectivities of a selection of electrophilic traps combined with established peptidomimetic sequences for the uPA, CatS, *β*5-subunit of the proteasome, SARS-CoV-2 M^pro^ and rhodesain, which represent cysteine, serine and threonine proteases. Based on these peptidomimetic sequences, we synthesized compounds decorated with warheads of different specificities. We chose the Michael acceptors ((F-)vinyl sulfone, nitroalkene and 4-oxoenoate) as cysteine-targeting and *β*-lactam as serine/threonine-targeting representatives. Furthermore, nitriles and *α*-ketobenzothiazoles were used, as they are applicable for both hydroxy- and thiol-containing nucleophiles. The compounds were tested on each target to analyze their affinities as well as their selectivity profiles.

Based on the in vitro studies, it is evident that the peptidomimetic sequences of the synthesized compounds play a crucial role in the selectivity towards the tested on-target and off-target proteases. This could be observed by the selectivity profile towards the cysteine protease M^pro^ and serine protease uPA. Only the inhibitors with the suited peptidomimetic sequence for M^pro^ (**84**, **88**, **90**, **93** and **94**) and for uPA (**103**) displayed inhibitory activity towards their targeted protease. Furthermore, the selection of a suitable warhead for the specific type of protease nucleophile ensures high affinity to the target or even activity in the first place, as demonstrated with the bortezomib congeners and the *α*-ketobenzothiazole inhibitor **103** as the only affine compound towards the uPA. The structurally similar papain-like proteases CatS and rhodesain showed that cross reactivity can occur, despite the design of well-defined peptidomimetic sequences. Therefore, the combination of both a highly reactive warhead towards the target protease, for example, the nitrile **30** group for CatS or the nitroalkene **13** for rhodesain, with a suitable peptidomimetic sequence can lead to potent inhibitors with promising pharmacodynamic properties.

Non-covalent docking yielded reasonable binding modes for all compounds resembling interactions of the crystallographic reference ligands and peptide recognition sequences in their expected subpockets. Additionally, electrophilic warheads were regularly found in close proximity to the nucleophilic catalytic amino acids, except for the *β*-lactams.

A reactivity test system with tool compounds of the used warheads and model nucleophiles was established to evaluate chemoselectivity. The findings confirmed the high reactivity of the 4-oxoenoate, the (F-)vinyl sulfones and the nitroalkene moieties towards the deprotonated thiol nucleophile/cysteine model, and high affinity of the Michael acceptor inhibitors towards the cysteine proteases was observed. Analogously to the in vitro studies of the uPA and M^pro^ target, the *α*-ketobenzothiazole warhead was found to be a potent electrophilic trap for both cysteine and serine proteases. Nevertheless, some major differences in reactivity could be observed, which might have been due to different conditions used in the chemical test system and the biochemical in vitro studies. To the best of our knowledge, this is the first extensive study in which different warhead types were combined with different peptidic recognition units and in which the resulting compounds were cross tested against different protease types. Similar published studies limited their focus to testing different warheads on one target or exchanging the peptidic backbone while retaining the same warhead [37,59,67,68].

## 4. Material and Methods

The material as well as the methods used for this study are described in the Supporting Information. The authors have cited additional references within the Supporting Information [21,42,43,46,47,63,64,69,70,71,72,73,74,75,76,77,78,79,80,81,82,83,84,85,86,87,88,89] (Appendix A of the reactivity study (Appendix A), of the fluorometric inhibition assays (Appendix A), of molecular docking (Appendix A), of quantum mechanics simulation (Appendix A) and of the NMR-spectra and HPLC-chromatograms of the final inhibitors (Appendix A) can be accessed in the supporting information).

## Data Availability

Not applicable.

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
