# Peer review of "Investigation of the Compatibility between Warheads and Peptidomimetic Sequences of Protease Inhibitors—A Comprehensive Reactivity and Selectivity Study"

_ijms, 2023, doi:10.3390/ijms24087226_

Round 1

Reviewer 1 Report

In this paper, the authors selected 7 well-know warheads and combined them with peptidomimetic sequences suited for five different proteases to provide a series of protease covalent inhibitors. Furthermore the resulting compounds were cross tested against different protease types, and molecular docking was performed to give insights into predicted binding modes of the inhibitors inside the binding pockets of the different enzymes. The reactivity of these warheads were investigated by NMR and LC-MS assays against serine/threonine and cysteine nucleophile models, as well as by quantum mechanics simulations. The workload of this work is very large.

 1.The research purpose of this work is not clear. From the field of medicinal chemistry, it is not meaningful to choose the substrates of other proteases and add warheads merely to determine the inhibition activity of the designated protease. That is, the random combination of warheads and peptidomimetic sequences cannot provide a clear conclusion. It is recommended that the authors delete unnecessary research and focus on studying warheads or peptidic backbones based on the specific structural characteristics of the target.

2. The numbering of some compounds in the Supporting Information section is confusing, for example the number of 2-phenylethanethiol.

3. In the SI section, it is necessary to provide spectra of all target compounds, not just spectrum analysis.

Author Response

Dear Reviewer 1:

Thank you for your revision.

To address your first point, we didn’t choose substrates randomly and combined them with warheads. The used peptidomimetic sequences were selected for the respective target protease specifically. To ensure a high suitability towards the protease of interest, we did intensive literature research (page 3-4, lines 122-146). Afterwards, we collected information about different kinds of warheads, regarding their electrophilic properties and inhibition mechanisms (reversible and irreversible), to get a well-balanced assortment for potentially targeting cysteine and serine/threonine proteases (page 3, lines 88-96).

We tested the synthesized compounds towards their target protease first, to investigate the affinity and reactivity properties of the different warheads, while retaining the suited peptidomimetic sequence, towards the specific protease and catalytic active amino acid. Afterwards, we did off-target studies with the respective inhibitor series towards the other proteases to determine the impact of the peptidomimetic sequence on the selectivity.

Based on the in vitro results and observed inhibitory activity, it is evident that the choice of the peptidomimetic sequence to the specific protease and the warhead-set is obviously meaningful. We could observe and evaluate the preference of the warheads towards certain proteases (e.g., α-ketobenzothiazole warhead towards the uPA, Figure 6 A). Additionally, the results indicated, that only the suited peptidomimetic sequence in combination with the α-ketobenzothiazole showed a productive enzyme inhibition towards uPA. In case of the Mpro we observed a high specificity of the suited peptidomimetic sequence for the target protease, since no inhibition was detected with other sequences (inhibition against Mpro could only be observed with the designed sequence, Figure 6 D). We could show that even with a suited peptidomimetic sequence, cross-reactivity can occur due to the selection of structurally similar proteases (see CatS vs rhodesain in vitro results Figure 6 C and Figure 6 E). By combining the two, a suited sequence plus a reactive warhead, it was still possible to design potent inhibitors with a selectivity advantage towards their off targets (page 27, lines 834-840). On the other hand, no potent inhibitor was found in case of the proteasome despite the correct peptide sequence, since none of the warheads was efficient, (page 26, lines 831-834).

It is of great interest to get both, selective and high affinity inhibitors during the design of potential drugs against proteases, which are associated with diseases. To the best of our knowledge other studies only focused on investigation of either warhead replacement or off-target selectivities. As a result of this, we are convinced that our study should contain the investigation of both, the exchange of warheads and the peptidomimetic sequences, to get a complete overview of the pharmacodynamic properties.

To 2: We deleted the structure number of 2-phenylethanthiol to be less confusing in the SI.

To 3: We added the NMR-spectra and UV chromatograms of the tested compounds in the SI.

Thank you for your time and effort.

Best,

Tanja Schirmeister.

Reviewer 2 Report

The authors descrived the investigation of the compatibility between warheads and peptidomimetic sequences of protease inhibitors.

It is a nice study!

The authors performed not only reactivity tests of all synthesized compounds, but also computational studies of those imprementation.

And also, the in vitro tests of 5 proteases with all synthesized compounds should be very interesting for many readers.

One thing to comment to the authors is, to assess time course of in vitro tests would be more useful for understanding feasibility of binding models with speculating Koffs.

But this manuscript could be worth to publicating in this journal. 

Author Response

Dear Reviewer 2:

Thank you for your revision. Unfortunately, we couldn’t understand what you were referring to with your suggestion. We think a more detailed look into binding kinetics would surpass our aim and we would like to conclude this study as it is.

Thank you for your time and effort.

Best,

Tanja Schirmeister.

Reviewer 3 Report

The article " Investigation of the compatibility between warheads and  peptidomimetic sequences of protease inhibitors - a  comprehensive reactivity and selectivity study is a good work; However, please note the following points for revision consideration:

1. The introduction section should benefit from a more explicit statement of the specific research question or hypothesis the study aims to address.

2  How does dysregulation of proteases contribute to severe pathologic conditions such as cancer, neurodegenerative or cardiovascular disorders?

3. How do covalent protease inhibitors function, and what are their advantages over non-covalent inhibitors?

4. How does changing the warhead affect a peptidomimetic inhibitor's reactivity and affinity towards a specific target protease?

5. What types of warheads can target thiol or hydroxy groups of amino acid residues?

6. What is the peptidic or peptidomimetic recognition sequence responsible for in the non-covalent interactions with substrate binding pockets, and how does it affect the selectivity profile of an inhibitor towards the protease of interest?

7. What is the impact of peptidomimetic sequences and different warheads on the affinity and selectivity towards selected target proteases, as assessed by in vitro testing, reactivity tests with model nucleophiles, and in silico studies?

8.The conclusion does not provide a clear and concise summary of the findings, making it difficult for the reader to quickly understand the main takeaways from the study.

Author Response

Dear Reviewer 3:

Thank you for your revision. In the following we will address each point individually. Newly written parts of the manuscript are marked in yellow.

To 1. The introduction was edited thoroughly for the following questions.

To 2. The Introduction already states the pathologic relevance of proteases (page 2, lines 39-50). For more in-depth information concerning the mechanisms of the pathologic conditions caused by the different proteases, we encourage the reader to have a look on the cited literature, since these mechanisms are already widely known and were described numerous times in the literature.

To 3. Covalent reversible and irreversible reactivity of inhibitors is already subject of the following section: page 2, lines 54-68.

Additionally, we added an explanation on why covalent-reversible inhibitors are preferred over irreversible ones (page 2, lines 67-68).

To 4. We integrated this question into our introduction, since this is one of the fundamental questions we aimed to investigate (page 3, lines 93-97).

To 5. This point has been addressed in the introduction (page 3, lines 78-87).

To 6. We addressed this in the introduction (page: 2, lines 69-71) and described it more in detail (page 2, line 71-73).

To 7. The results of the in vitro testing regarding the impact of the peptidomimetic sequence and the use of different warheads were discussed in the conclusion (page 26, lines 826-828) and were more precisely described exemplarily for the proteasome, Mpro and the uPA (modified: page 26, lines: 828-834).

To 8. We tried to present the received results of our study clearer, by mentioning specific examples, which highlight the gained conclusion of the peptidomimetic sequences, warheads, and combination of both. (modified: page 26, lines: 828-834).

Thank you for your time and effort.

Best,

Tanja Schirmeister.

Round 2

Reviewer 1 Report

All issues have been supplemented and corrected.